# Performance of arsenene and antimonene double-gate MOSFETs from first principles

Giovanni Pizzi[1,*], Marco Gibertini[1,*], Elias Dib[2,*], Nicola Marzari[1], Giuseppe Iannaccone[2] & Gianluca Fiori[2]

In the race towards high-performance ultra-scaled devices, two-dimensional materials offer an alternative paradigm thanks to their atomic thickness suppressing short-channel effects. It is thus urgent to study the most promising candidates in realistic configurations, and here we present detailed multiscale simulations of field-effect transistors based on arsenene and antimonene monolayers as channels. The accuracy of first-principles approaches in describing electronic properties is combined with the efficiency of tight-binding Hamiltonians based on maximally localized Wannier functions to compute the transport properties of the devices. These simulations provide for the first time estimates on the upper limits for the electron and hole mobilities in the Takagi's approximation, including spin–orbit and multi-valley effects, and demonstrate that ultra-scaled devices in the sub-10-nm scale show a performance that is compliant with industry requirements.

[1] Theory and Simulation of Materials (THEOS) and National Centre for Computational Design and Discovery of Novel Materials (MARVEL), École Polytechnique Fédérale de Lausanne, Lausanne CH-1015, Switzerland. [2] Dipartimento di Ingegneria dell'Informazione, University of Pisa, Pisa 56122, Italy. * These authors contributed equally to this work. Correspondence and requests for materials should be addressed to G.F. (email: gfiori@mercurio.iet.unipi.it).

In the past decades the exponential increase in computing power predicted by Moore's law has been enabled by scaling complementary metal-oxide-semiconductor silicon-based devices, that is, reducing their size and limiting at the same time their power dissipation, while increasing the operating frequency. With transistor dimensions going below 10 nm, fundamental limitations are emerging both in terms of manufacturing costs and device performance. To sustain Moore's law, a paradigm shift either in device architecture or in materials is needed.

In this respect, using two-dimensional (2D) systems as conduction channels is definitely one of the most exciting opportunities[1]. Indeed, their ultimate thinness can reduce short-channel effects, one of the main detrimental factors for devices at ultrashort lengths. For this reason, starting with the experimental realization of graphene[2], single-layer materials have gained considerable attention for a large number of different applications. Several studies, in particular, have targeted graphene as a component of novel devices, motivated by its exciting electronic, mechanical and thermal properties, such as its extremely high mobility[3]. Despite its appeal, graphene has regrettably no gap. Therefore, it is not suited for electronic applications such as field-effect transistors (FETs), where a semiconductor material with a finite gap is required for device switching. The first suitable candidate, the transition-metal dichalcogenide $MoS_2$ (ref. 4), has been shown to be an interesting transistor material, even if its mobility is much lower than that of graphene[5]. The list of relevant 2D systems has then been enriched by other transition-metal dichalcogenides[6] and by many other layered materials, such as black phosphorus and its monolayer form phosphorene, that is promising for its high mobility[7–12].

In light of the current pace at which 2D materials are being identified, we cannot expect that each new candidate is grown experimentally with high quality and then devices with various geometries are fabricated, characterized and optimized. Instead, simulations can be used to efficiently determine and optimize materials properties and device characteristics and filter only a few systems to send then to the laboratory. Promising candidates can be considered, for instance, by looking for materials chemically similar to existing ones. As an example, two new monolayer materials composed of group-V elements (in analogy with phosphorene) have been recently theoretically investigated: arsenene and antimonene[13–16], made of As and Sb, respectively. The authors have put forward the hypothesis that they could be attractive for device applications. While this suggestion is reasonable, only an accurate simulation of a complete device can support this hypothesis and will further stimulate experimental interest[17] in these novel 2D materials.

This task is not straightforward, however, because the simulation of a device requires a preliminary characterization of the material. While properties and parameters are available in the literature for well-studied systems (such as bulk Si or III–V semiconductors), in the case of new materials they are typically not available, nor they can be easily extracted from known systems; they must instead be calculated from first principles. This can be true even in simple cases: for instance, despite their chemical similarity, arsenene and antimonene display very different electronic and mechanical properties with respect to phosphorene, as they originate from different allotropes and have thus a completely different crystal structure. On the other hand, performing a full-device simulation directly from first principles is computationally out of reach. Device simulators based on effective tight-binding Hamiltonians[18,19] are viable, but require the knowledge of on-site and hopping energies, and a few different methods have been proposed in the literature to address the issue of bridging the different simulation scales[20,21].

Here, we adopt a multiscale approach based on maximally localized Wannier functions (MLWF)[22], while providing a physical understanding of the transport properties of monolayer As and Sb. Basic electronic properties are calculated from first principles using density-functional theory (DFT). The electronic wavefunctions are then used as input to obtain MLWF in a multiscale approach, providing us with an effective tight-binding Hamiltonian for the relevant electronic bands around the fundamental gap, and retaining at the same time full first-principles accuracy in the results[23,24]. MLWF are used to characterize the material (for example, effective masses) by exploiting the efficient band interpolation, and as a localized tight-binding basis set to simulate the currents in a complete device with a non-equilibrium Green function formalism[25]. In particular, we consider double-gate metal-oxide-semiconductor FETs (MOSFETs) based on arsenene and antimonene channels and compare their performance against industry requirements. We show that such devices have the potential to achieve the target set by the International Technology Roadmap for Semiconductors (ITRS)[26] for future competitive devices for high-performance digital applications, in particular in terms of the capability of behaving as an outstanding switch even in the ultra-scaled regime.

## Results

**Multiscale material characterization.** The first step towards the multiscale simulation of devices based on As and Sb monolayers is the computation from first principles of their electronic structure. To perform this task, we carried out DFT simulations using the Quantum ESPRESSO[27] suite of codes, efficiently automated using AiiDA[28] (more details in the Methods section). All calculations reported here include spin–orbit coupling (SOC) effects. In the Supplementary Note 1 we discuss in detail the effect of SOC and compare the results obtained here with those without SOC.

In Fig. 1a, we show the equilibrium crystal structure of arsenene and antimonene. Differently to phosphorene, As and Sb monolayers are not puckered, but display a buckled structure more similar to silicene or germanene[29,30], with two inequivalent atoms inside the primitive hexagonal unit cell lying on two different planes. (Note that As and Sb have also been predicted to exist in a puckered structure, but this phase is energetically less favourable[13,14]). The separation $d$ between the planes reads 1.394 Å for As and 1.640 Å for Sb, while the equilibrium lattice constant $a$ is, respectively, 3.601 and 4.122 Å. The band structure of both materials is very similar, and in Fig. 1b we show the energy bands along a high-symmetry path in the Brillouin zone obtained from DFT (empty circles) in the case of arsenene (for the bands of antimonene see Supplementary Fig. 1). The DFT bandgap is indirect (for both materials) and equal to 1.48 eV for As and 1.00 eV for Sb. The maximum of the valence bands is located at the $\Gamma$ point and, without SOC, it would be twofold degenerate; the inclusion of the SOC splits the degeneracy (see Fig. 1b and Supplementary Fig. 2) and the topmost valence band becomes non-degenerate, except for the twofold spin degeneracy. The minimum of the conduction bands lies instead along the $\Gamma - M$ line and gives rise to six valleys.

Further analyses of the electronic properties of arsenene and antimonene have been performed by first mapping the Bloch eigenstates associated with the bands around the gap into a set of maximally localized Wannier functions[22]. We focused in particular on the three top valence bands and three bottom conduction bands (per spin component). The main orbital contribution to these bands comes from $p$-orbitals of the atoms that form bonding and antibonding combinations around the gap. By projecting over the $p$-orbitals of the two atoms in the primitive cell, the standard localization procedure leads to six Wannier functions per spin

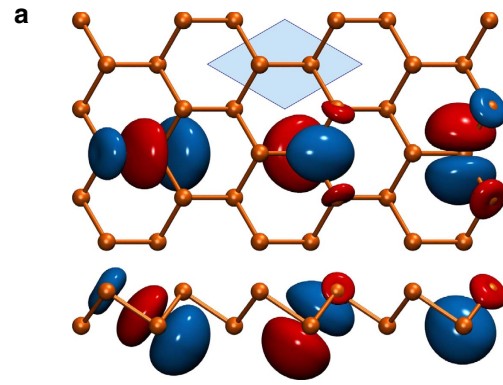

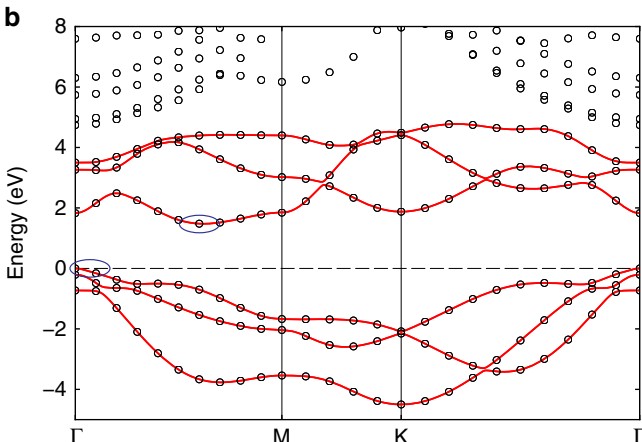

**Figure 1 | Crystal and electronic structure of As and Sb monolayers.**
(**a**) Top and lateral views of As or Sb monolayers. The blue-shaded region represents the primitive unit cell comprising two inequivalent atoms. The spatial profile of three maximally localized Wannier functions is also reported. Isosurfaces of different colours (red and blue) correspond to opposite values of the real-valued Wannier functions. (**b**) Energy bands of arsenene along a high-symmetry path in the Brillouin zone. Empty circles denote the results of a direct DFT calculation while red solid lines represent the Wannier-interpolated bands. Blue circles highlight the position of the valence band maximum and conduction band minimum.

**Table 1 | Valence and conduction effective masses of arsenene and antimonene.**

|  | As | Sb |
|---|---|---|
| $m_{DOS}^c$ | 0.270 | 0.261 |
| $m_L^c$ | 0.501 | 0.472 |
| $m_T^c$ | 0.146 | 0.144 |
| $m^v$ | 0.128 | 0.103 |

DOS, density-of-states; SOC, spin–orbit coupling.
Effective masses of the relevant bands of arsenene and antimonene, in units of the electron mass $m_0$, when SOC effects are included. Symbols are explained in the main text. Note that $m_{DOS}^c$ is the effective DOS mass for each of the six identical conduction band valleys.

topmost bands at $\Gamma$. Since the magnitude of the splitting is large, for realistic band filling levels we can limit ourselves to consider only the topmost valence band, with (isotropic) mass $m^v$. (If SOC was neglected, we would need instead to consider both degenerate bands, as discussed in Supplementary Notes 2–5 and Supplementary Figs 2–5.) For the conduction bands, the isoenergies of the six valleys are oblate with a larger effective mass $m_L^c$ in the longitudinal direction with respect to the transverse effective mass $m_T^c$. The effective DOS mass for each valley $m_{DOS}^c$ computed independently is in agreement with what can be expected from geometrical arguments, that is, $m_{DOS}^c \approx \sqrt{m_L^c m_T^c}$.

As a first assessment of the material properties toward the realization of a transistor device using arsenene or antimonene as channel materials, we estimate whether the ballistic approximation is valid at room temperature ($T = 300\,\text{K}$) in the ultra-scaled sub-10-nm regime that we investigate in this work. We will limit the analysis only to the intrinsic scattering with longitudinal acoustic phonons. As other scattering mechanisms may be active in the system, the values that we calculate should be considered as upper limits to the actual scattering times or, equivalently, to the carrier mobility. In particular, despite out-of-plane (ZA) phonons may play an important role in free-standing Dirac materials without planar symmetry[31,32], we do not consider them here. While in a free-standing material scattering with ZA phonons can be relevant, in our systems the device geometry (presence of substrate and of top gates) will shift the ZA phonon modes at finite energy, reducing their impact on the mobilities[33].

While accurate values for the electron–phonon scattering terms can be obtained fully *ab initio*[34,35], an efficient method to get estimates for the scattering times and mobilities relies on deformation-potential theory[36] and Fermi's golden rule to estimate the scattering times. An estimate of the 2D mobility can be then obtained using Takagi's formula[9,37,38]

$$\mu_{2D} = \frac{e\hbar^3 C_{2D}}{k_B T m_e^* m_{DOS}(E^i)^2},\tag{1}$$

where $m_e^*$ is the transport effective mass, $m_{DOS}$ the DOS effective mass (Table 1), $k_B$ the Boltzmann constant, $T$ the temperature, $E^i$ the deformation potential constant and $C_{2D}$ the elastic modulus. In our case, we need to consider this formula in the multi-valley, anisotropic case: details can be found in Supplementary Notes 3–9 and described in Supplementary Figs 3–7, as well as the values of the extracted relevant parameters (deformation potentials and elastic moduli, reported in Supplementary Tables 1 and 2, respectively).

We would like to emphasise, however, that this formula, while often adopted in the literature, cannot be used to obtain a quantitative estimate of the mobility. Indeed, the formula neglects the coupling with ZA phonons (which may be important, as already discussed above), as well as with TA and optical phonons.

component, three centred on one atom and three on the other. In Fig. 1a, we show the spatial profile of the three Wannier functions centred on atoms belonging to the lower plane (the other three can be obtained simply by spatial inversion through a mid-bond centre). They clearly have a *p*-like character with minor contributions from neighbouring atoms. From the knowledge of these Wannier functions it is straightforward to compute the matrix elements of the Hamiltonian between them.

In such a way, it becomes possible to interpolate efficiently the Hamiltonian at any arbitrary **k**-point in reciprocal space, keeping the same accuracy of the underlying first-principles simulation, but at an extremely reduced computational cost. In Fig. 1b, we show the Wannier-interpolated energy bands (red solid lines) with a much denser mesh than the original DFT results (empty circles) for As monolayer (bands for Sb monolayer are shown in Supplementary Fig. 1). Exploiting such interpolation scheme, we also computed the effective masses for relevant band extrema that crucially affect carrier mobilities and intraband tunnelling amplitudes. We both fitted the electronic bands along principal directions and evaluated accurately the density-of-states (DOS) on an extremely dense grid. The values of the masses are reported in Table 1 for both materials. In particular, for the valence band maxima, the SOC splits the degeneracy of the two

Moreover, it cannot fully capture the anisotropy of the electron–phonon coupling coefficients. A full *ab initio* treatment of the electron–phonon scattering is required, if a quantitative estimation is needed (see for example, discussions in refs 31,39). Nevertheless, we provide here an estimate of what we will call hereafter Takagi's mobility, mainly to allow to compare As and Sb with other 2D materials already investigated in the literature within the same level of theory. We have estimated that, in the worst-case scenario, the values of actual mobilities could be reduced up to a factor of 8 when a full treatment of the electron–phonon coupling is adopted, including intervalley scattering.

The resulting values of the electron Takagi's mobility $\mu_c$ and the hole Takagi's mobility $\mu_h$ are 635 and 1,700 $cm^2 V^{-1} s^{-1}$, respectively, for As and 630 and 1,737 $cm^2 V^{-1} s^{-1}$ for Sb.

The values of the electron Takagi's mobility are quite promising and comparable with theoretical results for phosphorene using the same level of theory[9] and even better than $MoS_2$ (ref. 40) owing to the smaller deformation potential. The hole Takagi's mobility is even larger, and in particular much larger than the experimentally measured value of the mobility in $MoS_2$ at room temperature[41] and in other 2D materials, like phosphorene[8]. On the other hand, we note that our predicted values are smaller than those predicted by simulations at the same level of theory (Takagi's formula) for phosphorene[9], owing to the larger elastic modulus and smaller deformation potential in the zigzag direction.

We also emphasise that in arsenene and antimonene the SOC effects are negligible for the conduction band. Instead, $\mu_h$ is significantly enhanced by the SOC, due to the splitting of the topmost valence band and the resulting changes in the effective masses and deformation potentials (see Supplementary Tables 1–3). In particular, the inclusion of the SOC increases $\mu_h$ by 25 and 84% in As and Sb, respectively (see Supplementary Table 4).

From these values of the Takagi's mobilities and the associated scattering times and carrier velocities reported in the Supplementary Note 10, we estimate that the mean free path limited by longitudinal acoustic phonons is of the order of tens of nanometre. For this reason we assume that, for the dimensions considered in this work, the use of the ballistic approximation is justified and sets a higher limit to the performance achievable in these devices.

**Performance of arsenene and antimonene based devices**. In view of the results of the previous section, we perform a full-device simulation of FETs based on arsenene and antimonene as channel materials; we will focus in particular on n-type devices since, as shown in Supplementary Note 10, they show better performance as compared with p-type FETs. Our aim is to assess quantitatively whether such devices can comply with industry requirements for high-performance applications as needed by the ITRS[26], which sets electrical and geometrical device parameters to keep the pace with Moore's Law[42]. The simulated device structure is shown in Fig. 2. We consider a double-gate transistor with doped source and drain, $SiO_2$ as gate dielectric and gate lengths ranging from 5 to 7 nm. The supply voltage ($V_{DD}$) and the oxide thickness ($t_{ox}$) are chosen according to the device channel length ($L_G$), as specified by ITRS. Spin–orbit coupling has been taken into account.

Figure 3 shows the transfer characteristics of As- and Sb-based MOSFETs for the set of parameters listed in Table 2. For a fair comparison, the gate work function of all devices has been shifted in order to have the same off-current $I_{OFF} = 0.1 A m^{-1}$ at $V_{GS} = V_{OFF} = 0 V$, that is, the smallest current driven by the transistor. Both As and Sb transistors show similar $I - V$ characteristics as a consequence of their very similar conduction bands[15].

From the $I - V$ characteristics, we can extract the main figures of merit required to assess the device performance, that we summarize in Table 2. In particular, the subthreshold swing (SS), defined as the inverse slope of the $I - V$ curve in semi-logarithmic scale in the subthreshold regime, provides relevant information regarding the sensitiveness of the device to short-channel effects: the smallest SS achievable in thermionic devices at room temperature is equal to 60 mV dec$^{-1}$ (ref. 43). For a gate length of 7 nm, both As and Sb based MOSFETs exhibit excellent SS: 64 and 60 mV dec$^{-1}$, respectively. As the channel length gets shorter ($L_G = 6$ and 5 nm), SS increases to 81 and 106 mV dec$^{-1}$ for As, and 83 and 106 mV dec$^{-1}$ for Sb transistors, respectively. The reported values of SS for both materials show very promising performances, maintaining a subthreshold slope of ∼100 mV dec$^{-1}$ even for the smallest devices.

Another figure of merit is the $I_{ON}$, that is, the largest current driven by the transistor (for $V_{GS} = V_{DS} = V_{DD}$.) All our considered devices comply with $I_{ON}$ requirements from ITRS. It is important to say that in our calculation the contact resistance has been neglected, and therefore our results represent an upper limit for the achievable $I_{ON}$.

The intrinsic delay time $\tau$ and the dynamic power indicator provide instead information regarding the switching speed and the power consumption of a device, respectively. The values we obtain comply with ITRS requirements, even for the shortest gate length. In particular, dynamic power indicator and $\tau$ are the energy and the time it takes to switch a complementary metal-oxide-semiconductor NOT port from the logic 1 to the logic 0 and vice versa, respectively. In the same Table 2, we also

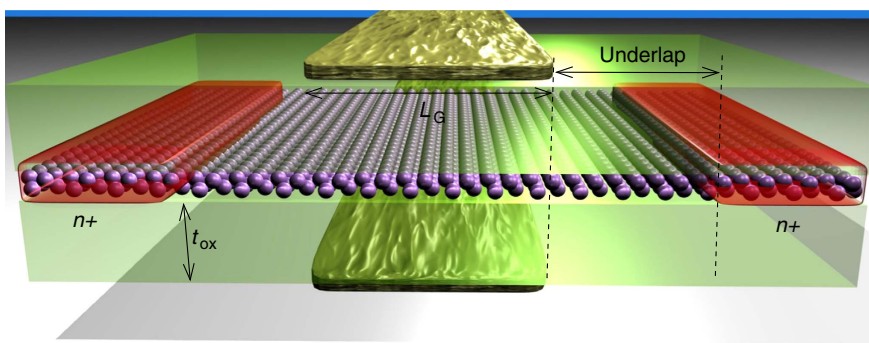

**Figure 2 | Double-gate n-doped MOSFETs.** Schematic view of the double-gate n-doped MOSFETs studied here, where the channel is either an As or an Sb monolayer. In the figure, the doped contacts, the gate and the oxide are shown, together with the main geometrical parameters of the device.

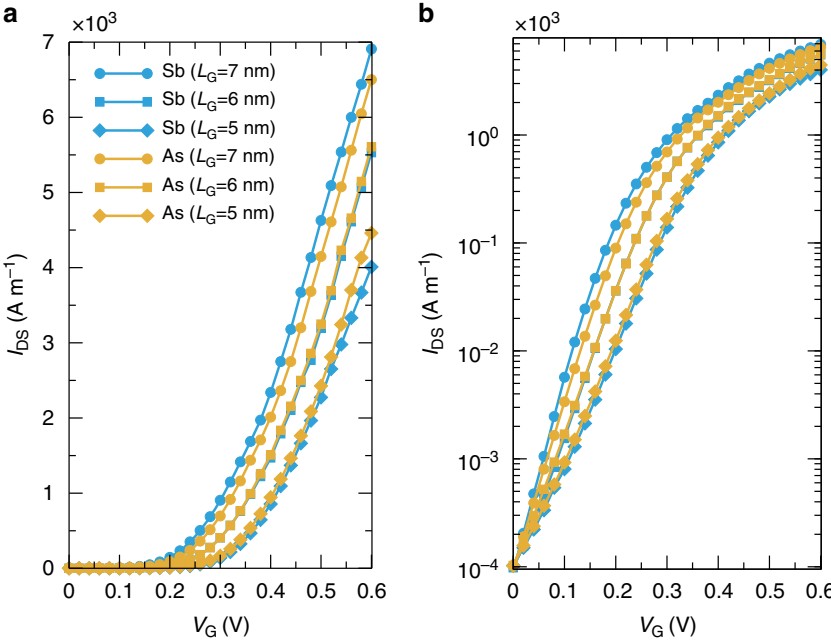

**Figure 3 | Transfer characteristics of As- and Sb-based MOSFETs.** $I_{DS} - V_{GS}$ curve in (**a**) linear and (**b**) semi-logarithmic scale for Sb (light-blue lines) and As (yellow lines) transistors with $L_G = 7$ nm, $V_{DS} = 0.6$ V and $t_{ox} = 0.5$ nm (circles), $L_G = 6$ nm, $V_{DS} = 0.57$ V and $t_{ox} = 0.45$ nm (squares) and $L_G = 5$ nm, $V_{DS} = 0.54$ V and $t_{ox} = 0.42$ nm (diamonds).

**Table 2 | Performance of arsenene and antimonene n-MOSFETs.**

| | | As | | | Sb | | |
|---|---|---|---|---|---|---|---|
| $L_G$ (nm) | | 7 | 6 | 5 | 7 | 6 | 5 |
| $t_{ox}$ (nm) | | 0.5 | 0.45 | 0.42 | 0.5 | 0.45 | 0.42 |
| $V_{DD}$ (V) | | 0.6 | 0.57 | 0.54 | 0.6 | 0.57 | 0.54 |
| SS (mV dec$^{-1}$) | This work | 64 | 81 | 106 | 60 | 83 | 106 |
| $I_{ON}$ (A m$^{-1}$) | (ITRS) | $\geq 2.19 \times 10^3$ | $\geq 2.31 \times 10^3$ | $\geq 2.41 \times 10^3$ | $\geq 2.19 \times 10^3$ | $\geq 2.31 \times 10^3$ | $\geq 2.41 \times 10^3$ |
| | This work | $6.57 \times 10^3$ | $4.9 \times 10^3$ | $3.2 \times 10^3$ | $6.91 \times 10^3$ | $4.93 \times 10^3$ | $2.98 \times 10^3$ |
| $\tau$ (ps) | (ITRS) | $\leq 0.125$ | $\leq 0.1$ | $\leq 0.08$ | $\leq 0.125$ | $\leq 0.1$ | $\leq 0.08$ |
| | This work | 0.04 | 0.045 | 0.052 | 0.042 | 0.047 | 0.055 |
| $f_T$ (THz) | (ITRS) | $\geq 1.91$ | $\geq 2.36$ | $\geq 2.88$ | $\geq 1.91$ | $\geq 2.36$ | $\geq 2.88$ |
| | This work | 5.8 | 6.01 | 5.47 | 5.51 | 5.94 | 4.83 |
| DPI ($10^{-10}$ J m$^{-1}$) | (ITRS) | $\leq 1.6$ | $\leq 1.4$ | $\leq 1.2$ | $\leq 1.6$ | $\leq 1.4$ | $\leq 1.2$ |
| | This work | 1.58 | 1.25 | 0.91 | 1.76 | 1.31 | 0.89 |

DPI, dynamic power indicator; FET, field-effect transistor; FOM, figures of merit; ITRS, International Technology Roadmap for Semiconductors; MOSFET, metal-oxide-semiconductor FETs.
Device parameters and calculated figures of merit of As- and Sb-based n-MOSFETs for different channel lengths. The target FOMs set by ITRS for end-of-the-roadmap are also indicated[26]. The meaning of each parameter is explained in the main text.

show the cutoff frequency $f_T$, that is, the frequency for which the current gain of the transistor is equal to one, which is a relevant parameter for radio-frequency applications. Both As and Sb MOSFETs exhibit excellent $f_T$ compared with ITRS. As compared with other 2D materials, As and Sb show performance comparable to that obtained in black phosphorus FETs[44,45].

To get a deeper understanding of the effects limiting the device performance, we focus in particular on the degraded SS observed in short-channel devices, which can be attributed to two main phenomena: large tunnelling currents through the narrow barrier; and large parasitic capacitance at source/drain-channel junctions, that is, short-channel effects. To elucidate which of the two effects plays a major role, we consider them separately for the shortest device: we either neglect quantum phenomena for the current (that is, tunnelling through the channel barrier), but not when computing the charge (that is, we consider mid-gap tunnelling states when solving the electrostatic problem, red line in Fig. 4) or

vice versa (yellow line in Fig. 4). Transfer characteristics with almost ideal SS (60 mV dec$^{-1}$) are obtained in the first case, demonstrating that the SS in short-channel devices is limited by the fact that the channel barrier is almost transparent for electrons injected from the source reservoir, and not by the short-channel effects, as one may expect for such short-channel lengths. This suggests that, from an engineering point of view, in order to improve the performance in ultra-scaled devices, efforts have to be directed in increasing the opacity of the channel barrier. This can be achieved for example exploiting materials with larger longitudinal tunnelling effective masses, or using uniaxial strain to split the conduction valleys while selecting bands with larger tunnelling effective mass in the transport direction.

Performing an investigation along the device parameter space, we have also computed the $I - V$ characteristics for different gate underlap values (defined in Fig. 2), fixing the distance

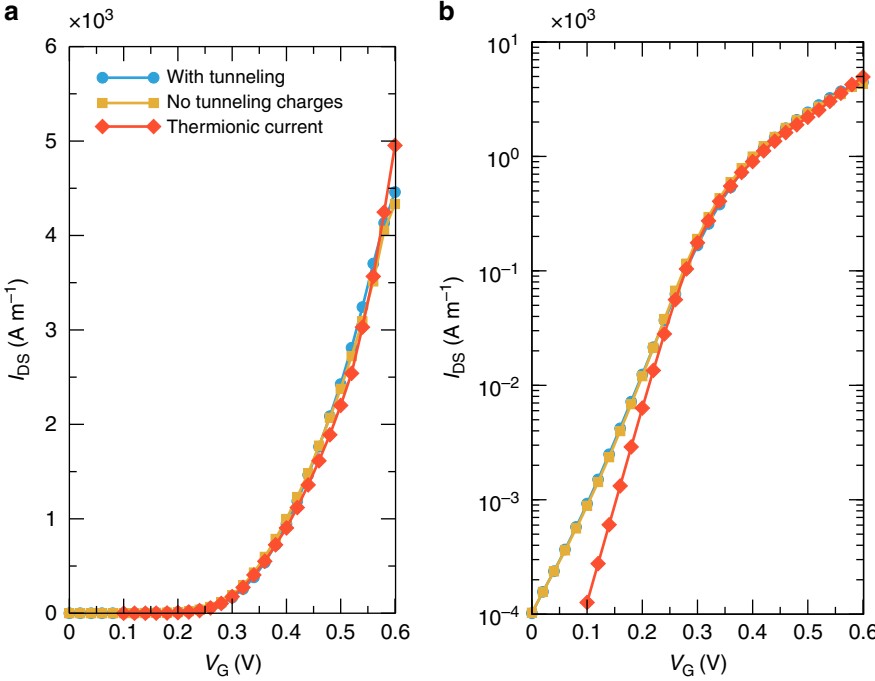

**Figure 4 | Short-channel effects on subthreshold swing.** $I_{DS} - V_{GS}$ curve in (**a**) linear and (**b**) semi-logarithmic scale for As transistors with $L_G = 5$ nm and $V_{DS} = 0.54$ V. The full-simulation model considering tunnelling both in the current and the charges is represented by light-blue circles, whereas suppressed charges in the channel and thermionic currents only are represented by yellow squares and red diamonds, respectively.

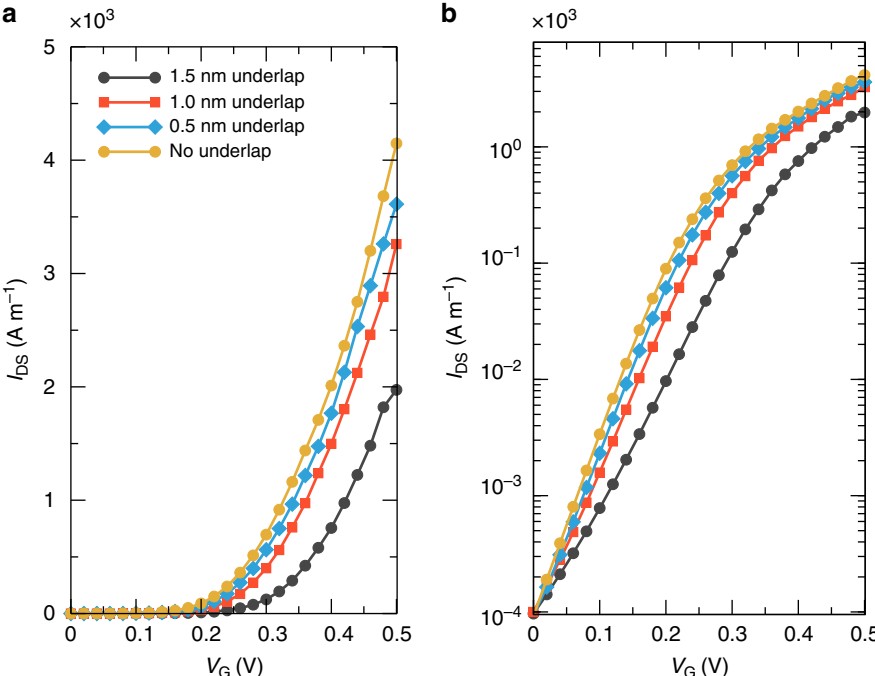

**Figure 5 | Transfer characteristics for different underlap.** $I_{DS} - V_{GS}$ curve in (**a**) linear and (**b**) semi-logarithmic scale for As transistors with total distance between drain and source electrodes of 7 nm, and different underlap.

between source and drain electrodes (that is, 7 nm) and changing accordingly the gate length $L_G$ and the underlap region, and the source and drain doping concentrations (Figs 5 and 6, respectively). As it can be seen from the results reported in Fig. 5, $I - V$ curves change only marginally when considering different underlap values. As a consequence, from a fabrication point of view, while control of the geometrical parameters for the gate contacts is required, minor dispersions do not drastically degrade the device performance. From Fig. 6, instead, we deduce that the subthreshold slope improves significantly when the doping is reduced. Therefore, the doping concentration can be used as an additional parameter to optimize the device performance.

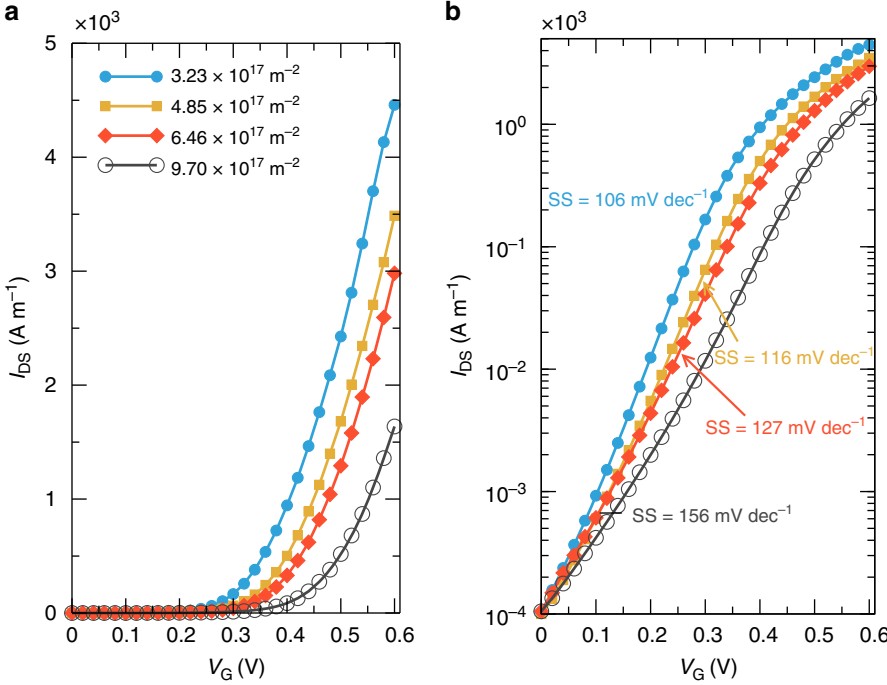

**Figure 6 | Boosting device performances by tuning source and drain doping concentrations.** $I_{DS} - V_{GS}$ curve in (**a**) linear and (**b**) semi-logarithmic scale for As transistors with $L_G = 5$ nm and for different source and drain doping concentrations.

## Discussion

In summary, we have provided a comprehensive analysis of 2D FET transistors based on arsenene and antimonene (that is, monolayers composed of As and Sb, respectively), demonstrating that these materials are promising for high-performance devices for digital applications. Our single-valley and multi-valley upper estimates of the mobilities in the Takagi's approximation show that high phonon-limited mobilities can potentially be obtained both in monolayer As and Sb, even if *ab initio* simulations of the electron–phonon scattering are required to obtain quantitative predictions for this quantity. This result motivated our extensive investigation of device performance using a multiscale approach, where the predictive power of DFT calculations has been incorporated into an efficient tight-binding model using maximally localized Wannier functions. When exploited as channel materials in FETs, arsenene and antimonene show a performance compliant with industry requirements for ultra-scaled channel lengths below 10 nm, where the ultimate atomic thickness of the exploited 2D materials effectively manages to suppress short channel effects and tunnelling starts to play a major role. We expect therefore that our predictions will provide a strong motivation for further experimental investigation of these novel materials.

## Methods

**First-principles calculations.** All first-principles calculations have been performed using DFT as implemented in the pw.x code of the Quantum ESPRESSO v.5.1.2 distribution[27], using the Perdew-Burke-Ernzerhof (PBE) exchange-correlation functional[46]. Pseudopotentials and energy cutoffs in a plane-wave basis have been chosen using the converged results provided in the SSSP pseudopotential library[47] for calculations without SOC. In particular, we used an ultrasoft pseudopotential[48] from PSLibrary[49] with cutoffs of 40 and 320 Ry (for the expansion of the wavefunctions and the charge density, respectively) for As; and an ultrasoft pseudopotential from the GBRV library[50] (with cutoffs of 50 and 400 Ry, respectively) for Sb. For calculations including SOC we used instead norm-conserving pseudopotentials from the Pseudo Dojo project[51] with cutoffs of 40 and 160 Ry (for the expansion of the wavefunctions and the charge density, respectively) for As; and with cutoffs of 80 and 320 Ry, respectively, for Sb.

Supercells with 20 Å of vacuum in the direction orthogonal to the layers have been considered to minimize the interaction between periodic replicas. Integrals on the Brillouin zone have been performed on a $14 \times 14 \times 1$ Γ-centred grid for the primitive cell (two atoms) and on a $6 \times 10 \times 1$ Γ-centred grid for the rectangular supercell (containing four atoms). Convergence on the charge density in the self-consistent loop was considered achieved when the estimated energy error was smaller than $1 \times 10^{-8}$ meV. Structures have been relaxed using the Broyden–Fletcher–Goldfarb–Shanno algorithm until forces were smaller than 1 meV Å$^{-1}$. The same parameters have been used also for relaxing atomic positions at fixed cell for the evaluation of elastic moduli and deformation potentials.

**Wannier functions.** MLWF[22,52,53] have been computed using v.2.1 of the Wannier90 code[54,55]. The same k-grids used for the computation of the DFT charge densities have been employed to compute the wavefunctions and overlap matrices used as input to calculate Wannier functions. The lower-energy bands have been explicitly excluded from the calculation: with the pseudopotentials we used, 2 for As and 12 for Sb in the primitive cell (4 and 24, respectively, when considering explicitly the spin degeneracy), and only six bands (12 with spin degeneracy) around the fundamental gap have been considered. We have chosen p-type orbitals centred on each atom in the cell as initial projections. Convergence has been considered achieved when the change in the total spread was smaller than $10^{-12}$ Å$^2$ for at least 20 iterations. Wannier functions have then been used to compute band structures and DOS on denser k-grids ($400 \times 700$ in the rectangular cell). DFT and Wannier calculations have been managed using the AiiDA framework[28] v. 0.5.0 to manage, automate and store in a graph database calculations, results, and computational workflows (for example, for band structure calculations, Wannierization, effective mass evaluations).

**Device simulations.** The Hamiltonian expressed on the MLWF basis set has been exploited in order to compute transport within the non-equilibrium Green function formalism[25]. The system is considered infinite along the zigzag direction (with Bloch periodic boundary conditions), while the transport channel is along the armchair direction. To compute the currents in the device in the ballistic regime, we have used the open-source NanoTCAD ViDES (ref. 18) code. In particular, in order to accurately reproduce the energy bands obtained from first-principles, up to 58 nearest-neighbours have been included in the Hamiltonian, and transport problems have been solved considering 30 wavevectors in the Brillouin zone and an energy step of 1 meV. For the electrostatic problem, the 2D Poisson equation has been solved, while potential translational invariance has been considered in the direction transversal to transport. All transport calculations are performed at room temperature. In all device simulations (except where explicitly otherwise

mentioned) a doping concentration of $3.23 \times 10^{17} \, \mathrm{m}^{-2}$ has been considered for the source and drain contacts.

**Data availability.** The data that support the findings of this study are available from the corresponding author upon request.

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

## Acknowledgements

We thank Nicolas Mounet for providing us with the AiiDA workflow to compute band structures with Quantum ESPRESSO and Massimo Fischetti for very insightful discussions. This work was supported by a grant from the Swiss National Super-computing Centre (CSCS) under project IDs s580. M.G., N.M., G.I. and G.F. gratefully acknowledge the Graphene Flagship (contract 696656).

## Author contributions

G.P., M.G. and G.F. conceived the work. G.P. and M.G. performed first-principles and Wannier simulations under the supervision of N.M. E.D. performed the NEGF device simulations under the supervision of G.I. and G.F. All authors analysed the results and wrote the paper.

## Additional information

**Competing financial interests:** The authors declare no competing financial interests.

