## [Peer review file · Nature Communications]

Reviewers' comments:

Reviewer #1 (Remarks to the Author):

The authors calculate the performance of n-MOSFET based on arsenene and antimonene. Recently many exotic 2D materials have been explored. Arsenene and antimonene are one of the most recently predicted materials. The study of their applications are very important. Especially, it is interesting that the mobility is as high as phosphorene and much larger than TMDs. This paper has a potential to be published in Nature Communications. However the present manuscript has several shortcomings and is unsatisfactory. The authors should properly answer to the following comments.

1)

First of all, there is no information of the tight-binding model of arsenene and antimonene although the authors claim that they have constructed them by the maximally-localized Wannier functions method.

The authors should explicitly show the data so as to the readers can reproduce the results.

All the hopping and on-site energies for each orbitals are necessary.

Furthermore, the authors should show the components of 6 orbitals.

These information is vital for the publication of this paper.

2)

The authors should discuss more in details the difference between phosphorene, arsenene and antimonene.

For example, what material is best for n-MOSFET?

3)

With respect to the statement

"This can be true even in simple cases: for instance, despite their chemical similarity, arsenene and antimonene display very different electronic and mechanical properties with respect to phosphorene, as they originate from different allotropes and have thus a completely different crystal structure. "

There is also the puckered structure of arsenene as shown in Ref.13.

The authors should mention of this fact.

How do the results change if the structure is puckered?

4)

Arsenene and antimonene are indirect semiconductors.

It seems that direct semiconductors are superior than indirect semiconductors for MOSFET.

Needless to say, phosphorene is a direct gap semiconductor.

Is there no demerit of arsenene and antimonene since they have indirect gaps?

5)

What is the reason that arsenene and antimonene have large mobilities?

The authors should compare and discuss the mobilities between phosphorene, arsenene and antimonene.

What is the reason that the mobilities of these materials are much larger than TMDs?

6)

In realistic situation, there must be impurities, defects and edge randomness, which reduce the performance of MOSFET.

The authors should estimate these effects.

7)

Do the authors include the finite temperature effects?

It seems that the calculations are based on the zero-temperature.

However, it is important to study the performance at room-temperature.

Does the performance decrease at room temperature?

8)

In general, the results shown in figures are much similar between arsenene and antimonene

although the band structures are different.

What are the reasons of this coincidence?

9)

What can we learn from the effective mass shown in Table I?

What are the characteristic features of these masses compared with other materials such as phosphorene and TMDs?

10)

From Table II,

it seems that the performance is best for $LG=7$.

If so, it is expected that the performance becomes better for larger LG .

Why do not the authors calculate with larger LG ?

Does the performance becomes better and better for larger LG or is it saturated for large LG ?

11)

The authors should explain more in details the meanings of the Table II.

The readers cannot understand the meaning of Table II.

Reviewer #2 (Remarks to the Author):

in this manuscript the authors present a theoretical study of the electronic structure and electronic transport in phosphorene and antimonene and investigate the ballistic performance of nanometer-scale field-effect transistors (FETs) that employ these 2D materials as active transport channels. The major emphasis of the work is on the ability to perform multi-scale simulations, starting from the first-principle (density-functional theory, DFT) calculations of the band structure of these materials, moving to calibrated tight-binding (Wannier-based) models that are used to study the performance of ballistic FETs.

There can be no doubt that the subject is extremely interesting and timely. However, I do not think that the results presented here are sufficiently convincing. Indeed, whereas the methodology used by the authors is sound, too many severe additional physical effects have been ignored. In addition, a vast literature seems to have been ignored. Many of the results presented here seem to be minor extensions of work that has already been published. The authors should cite papers that are very relevant to the subject at hand (examples given below), consider the comments below, and resubmit their manuscript for further consideration.

To be specific:

1. The thermodynamic stability of 2D crystals is known to cause a strong coupling between electrons and flexural acoustic modes (ZA phonons). In "buckled" materials -- such as phosphorene and antimonene -- this coupling is first-order (unlike what happens in graphene) and causes potentially serious issues. This problem has been hinted recently by Gunst et al. *Phys. Rev. B* vol. 93, 0354141 (2016) and discussed at length by Fischetti et al., *ArXiv* 1511:0769v1 (1015). Yet, the authors make no mention of this issue that may have strong implications on electron transport. See also Roome et al., *Appl. Mat. Interf.* vol. 6, 7743 (2014).
2. It is not clear whether or not the spin-orbit interaction has been included in the DFT or tight-binding band structure calculations. Clearly, this is a major effect that affects the band structure and must be accounted for. After all, Sb monolayers are known to be 2D topological insulators thanks to this effect. The properties of the electron-phonon interaction (viz., deformation potentials) are strongly affected by the spin-orbit coupling.
3. The electron-phonon matrix element exhibits a strong anisotropy in many cases. It is not clear how this has been accounted for by the authors who consider simple deformation potentials obtained from the applied strain rather than from a direct calculation of the electron-phonon matrix elements, as it should be done.
4. The "Takagi" formula for the carrier mobility is notoriously oversimplified. It has been used previously (see Shao et al., *J. Appl. Phys.* vol. 114, 093712 (2013) for the case of silicene), but it neglects important wavefunction-overlap, degeneracy, and band-structure effects. See also van de Broek et al., *2D Mat.* vol. 3, 015001 (2016) for Sn, Si nanoribbons, for example.
5. The use of Wannier functions to "calibrate" tight-binding models has been described and used

recently by the Purdue group (Kubis, Klimeck et al.).

Reviewer #3 (Remarks to the Author):

A. Summary of key results

A very promising n-channel transistor design as a switch where channel (less than 10nm) is formed from high mobility 2D materials - arsenene and antimonene. The transistor outperforms widely referred 2D material - MoS₂. Multiscale simulation approach is used in device modeling. Short-channel effects are suppressed due to atomic layer thickness of materials (As, Sb). The comparison is made with the performance of ITRS for the targeted year 2026 n-MOSFETS and the results are very promising.

B. Originality and interest

Use of high mobility 2D atomic layer materials as a channel in the sub-10nm scale as compared to other materials such as MoS₂ in double-gate transistor structure and compliance with the industry requirements (ITRS 2026 nMOSFET performance) is an original work, novel and will generate major interest in this field.

C. Data and methodology

Authors have described bridging the different simulation tools and using the approach based on the first-principles methods typically used in studying electronic behavior of materials and extracting electrical performance in terms of subthreshold slope, on/off current, delay and cut-off frequency, etc. The method is described and all tools used both in Open Source code and commercial (AiiDA) are described.

D. Appropriate use of statistics and treatment of uncertainties

Device design and performance is based on simulation tools and their limitations are known. The results are reliable.

E. Conclusions: robustness, validity and reliability

The essence of the work is summarized in an integrated form and as stated in D, the results are reliable.

F. Suggested improvements: experiments, data for possible revision

It is left to authors' discretion to add a paragraph on how the 2D arsenene and antimonene based n-MOSFET can be realized experimentally.

G. References: appropriate credit to previous work?

Relevant references are cited with due credit to these references.

H. Clarity and context: lucidity of abstract/summary, appropriateness of abstract, introduction and conclusions

Some of long paragraphs in Introduction and Results & Discussion section are too long. It is advised to split in small paragraphs so that work will read more focused.

Detailed Reply to the Reviewers

Reviewer #1

- 1) First of all, there is no information of the tightbinding model of arsenene and antimonene although the authors claim that they have constructed them by the maximally localized Wannier functions method. The authors should explicitly show the data so as to the readers can reproduce the results. All the hopping and onsite energies for each orbitals are necessary. Furthermore, the authors should show the components of 6 orbitals. These information is vital for the publication of this paper.*

The tightbinding model for arsenene and antimonene has been obtained using the opensource Quantum ESPRESSO and Wannier90 codes, and all input parameters have been provided in the

Methods section. However, reproducing the calculations can require some time, therefore the Reviewer's suggestion of providing the tightbinding parameters of our simulations is well motivated. We attach therefore two files (As_soc.zip and Sb_soc.zip) with the Hamiltonian in real space between the MLWFs (i.e. the basis set of our tightbinding model). The format of the file is the one produced by Wannier90, described in section 8.18 (page 76) of the code documentation, available online at the following address: http://www.wannier.org/doc/user_guide.pdf – in particular, it contains the (complex) matrix elements between basis functions in a supercell. The coordinates of the orbitals, if needed, are reported at the end of the output file (.wout extension) of the code. The Hamiltonian can be used directly with the Wannier90 code without the need of changing the format, and with the NanoTCAD ViDES code, or with any other code just by converting the file format. If it is possible to include .zip files as Supplementary Material of the paper, we would be more than happy to add these files; in any case, we expect to open (within a couple of months) a section to our materialscloud.org portal that we are building, that will host all inputs and outputs of our published calculations.

2) *The authors should discuss more in details the difference between phosphorene, arsenene and antimonene. For example, what material is best for nMOSFET?*

The current work purposely focuses on new materials, i.e., As and Sb, while the study of the transport characteristics of phosphorene falls outside the scope of the current article. Phosphorene transistors have indeed been already investigated previously using numerical simulations (e.g., K. Lam et al., IEEE EDL 35, 963, (2014); F. Liu et al., IEEE Trans. Electr. Dev. 61, 3671 (2014)), showing performances close to the ones obtained for As and Sb based transistors. The main text has been modified including the two references and adding a comment comparing the investigated materials with black phosphorus: "As compared to other twodimensional materials, As and Sb show performance comparable to that obtained in black phosphorus FETs [REFS]".

3) *With respect to the statement "This can be true even in simple cases: for instance, despite their chemical similarity, arsenene and antimonene display very different electronic and mechanical properties with respect to phosphorene, as they originate from different allotropes and have thus a completely different crystal structure." There is also the puckered structure of arsenene as shown in Ref.13. The authors should mention of this fact. How do the results change if the structure is puckered?*

Indeed, As has also been predicted to exist in a puckered structure in Ref. 13. We have added a sentence in the beginning of the "Multiscale material characterisation" to mention this. We do not further investigate the puckered structure in the paper because we wanted to focus on realistic devices and on the most probable crystal structure for arsenene. Indeed, in Ref. 13 the puckered structure is shown to be energetically less favourable. Moreover, also in the parent 3D material, As is layered and the layers have a buckled structure, therefore the buckled structure is the easiest to obtain using exfoliation techniques. For what concerns the changes in the results for a puckered structure, we can predict that if such a structure could be realized, the results (for ntype devices) would be similar to those of phosphorene. This is motivated by the similarity of the conduction band structures of phosphorene (see e.g. Fig. 2 of J. Qiao et al., Nat. Commun. 5, 4475 (2014).) and puckered arsenene (Fig. 1 of Ref. C. Kamal, Phys. Rev. B 91, 085423 (2015)) (both references are cited in the paper). In the valence band, two bands become almost degenerate in the valence, and

therefore accurate calculations need to be performed for a quantitative estimate of the results, but this is beyond the scope of the present paper.

- 4) *Arsenene and antimonene are indirect semiconductors. It seems that direct semiconductors are superior than indirect semiconductors for MOSFET. Needless to say, phosphorene is a direct gap semiconductor. Is there no demerit of arsenene and antimonene since they have indirect gaps?*

Actually, direct bandgap plays an important role especially in applications exploiting interband tunneling (i.e., valence-to-conduction band tunneling) as for example in optoelectronic devices or tunnel FET transistors. In our specific case, we are considering unipolar thermionic devices, where interband tunneling is not relevant, so direct bandgap materials are not supposed to show better performance as compared to indirect bandgap materials.

- 5) *What is the reason that arsenene and antimonene have large mobilities? The authors should compare and discuss the mobilities between phosphorene, arsenene and antimonene. What is the reason that the mobilities of these materials are much larger than TMDs?*

Within our formalism the main factors affecting mobilities are: effective masses, deformation potentials, and elastic constants. Comparing arsenene and antimonene with phosphorene (J. Qiao et al., Nat Commun. 5, 4475 (2014)) one notes that effective masses are better (i.e. smaller) in our case, while deformation potentials and elastic constants are better for phosphorene, especially for holes along the zigzag direction. Overall this gives comparable mobilities for electrons, while hole mobilities are larger in phosphorene (although numerical results on the same level of theory are very noisy in J. Qiao et al., Nat. Commun. 5, 4475 (2014)). As far as TMDs are concerned, the better results that we obtain are related to both a smaller deformation potential for electrons and a smaller effective mass for holes, which both enter quadratically in the expression for the mobility. We have extended this analysis in the main text (with the appropriate references): “The values of the electron mobility are quite promising and comparable with theoretical results for phosphorene using the same level of theory and even better than MoS₂ owing to the smaller deformation potential. The hole mobility is even larger, and in particular much larger than the experimentally measured value in MoS₂ at room temperature and in other 2D materials, like e.g. phosphorene (although smaller than the values predicted by simulations at the same level of theory for phosphorene, owing to the larger elastic modulus and smaller deformation potential in the zigzag direction).”

- 6) *In realistic situation, there must be impurities, defects and edge randomness, which reduce the performance of MOSFET. The authors should estimate these effects.*

Our study aims to set a higher limit for the performances of As and Sb monolayer based nMOSFETs. In the best case scenario, i.e. when neglecting defects and edge roughness, we gave a theoretical prediction of the figures of merit neglecting experimental constraints as a first attempt to evaluate these newly predicted 2D materials. Indeed, a simple estimation of the effects of impurities and defects is not available, and an accurate simulation of these effects would require extensive calculations, beyond the scope of the present work.

7) *Do the authors include the finite temperature effects? It seems that the calculations are based on the zerotemperature. However, it is important to study the performance at room-temperature. Does the performance decrease at room temperature?*

Our transport calculations are performed at room temperature. Indeed, this was not mentioned explicitly in the paper, so we have added a sentence in the Methods section: “All transport calculations are performed at room temperature”. In particular, the effect of temperature leads to the broadening of the Fermi distribution function involved in the calculation of the current (through the LandauerButtiker formalism) and the charge density that modifies the electrostatic potential within the PoissonGreen solver, which in turn modifies the final current, once convergence is reached

8) *In general, the results shown in figures are much similar between arsenene and antimonene although the band structures are different. What are the reasons of this coincidence?*

The band structures of the two materials are, actually, very similar. The main difference between the two materials is the energy gap. This is visible, for instance, in the new Supplementary Figure 2 (black curves, where the spinorbit coupling is not included, as it was in the original version in the paper). The reason is due to the structural and chemical similarity between the two materials. Please note also that the inclusion of the spinorbit coupling (included in the revised manuscript) makes the band structures of the two materials a bit more different especially in the valence band (because the magnitude is different in the two materials). The conduction bands (and therefore the behavior of nMOSFET devices) are still very similar.

9) *What can we learn from the effective mass shown in Table I? What are the characteristic features of these masses compared with other materials such as phosphorene and TMDs?*

Effective masses have two important consequences in our analysis: (i) on the mobilities and (ii) on intraband tunneling. The smaller the effective masses, the larger mobilities are, but at the same time the larger the tunneling probability. In order to have good device characteristics one thus needs the right tradeoff and our multiscale approach offers an efficient method to assess these properties. For what concerns the comparison with phosphorene and TMDs we refer to the reply to point 5 above. In order to clarify this issue we have revised the manuscript as described in point 5.

10) *From Table II, it seems that the performance is best for LG=7. If so, it is expected that the performance becomes better for larger LG. Why do not the authors calculate with larger LG? Does the performance becomes better and better for larger LG or is it saturated for large LG?*

We agree that longer channels will result in better overall characteristics in the ballistic approximation. However, when evaluating the performances of MOSFETs, one also has to evaluate the scalability for a given channel material. Indeed, in order to comply with Moore's law the MOSFET dimensions need to be reduced at each technological node. As predicted by the ITRS, the gate lengths are expected to be below 7 nm at the endoftheroadmap, thus we simulated devices with these dimensions.

11) *The authors should explain more in details the meanings of the Table II. The readers cannot understand the meaning of Table II.*

We thank the Reviewer for her/his comments. The way the data is presented in Table II has been changed in order to improve the clarity of the quantities shown in the Table (e.g., indicating if the ITRS values should be considered as minimum or maximum requirements, and improving the overall layout). We have also added additional explanations of the physical meaning of each of the quantities in the main text: “ I_{ON} , i.e., the largest current driven by the transistor (for $V_{GS} = V_{DS} = V_{DD}$)”; “the cutoff frequency f_T , i.e., the frequency for which the current gain of the transistor is equal to one”; “ DPI and τ are the energy and the time it takes to switch a CMOS NOT port from the logic “1” to the logic “0” and viceversa, respectively.”

Reviewer #2

1. *The thermodynamic stability of 2D crystals is known to cause a strong coupling between electrons and flexural acoustic modes (ZA phonons). In "buckled" materials such as phosphorene and antimonene this coupling is firstorder (unlike what happens in graphene) and causes potentially serious issues. This problem has been hinted recently by Gunst et al. Phys. Rev. B vol. 93, 0354141 (2016) and discussed at length by Fischetti et al., ArXiv 1511:0769v1 (1015). Yet, the authors make no mention of this issue that may have strong implications on electron transport. See also Roome et al., Appl. Mat. Interf. vol. 6, 7743 (2014).*

We totally agree with the Reviewer that the absence of mirror symmetry in freestanding Dirac 2D materials might lead to a strong electronphonon coupling with outofplane phonons (ZA). On the other hand, arsenene and antimonene do not have a Dirac dispersion, but display a finite energy gap. In addition, we are interested in a device geometry in which the presence of a substrate and of a topgate breaks the translational invariance along the vertical direction, so that the ZA phonons exhibit a finite energy in the longwavelength limit. This greatly reduces their effect on the intrinsic transport properties of the devices under investigation. We thus believe that in our case ZA phonons do not play a major role and can be safely disregarded. We have added a sentence in the manuscript to address this issue, including the suggested references: “In particular, while outofplane (ZA) phonons may play an important role in freestanding Dirac materials without planar symmetry [REFS: Gunst,Fischetti], we do not consider them here. In fact, the device geometry (presence of substrate and of top gates) will shift the ZA phonon modes at finite energy, reducing their impact on the mobilities”.

2. *It is not clear whether or not the spinorbit interaction has been included in the DFT or tight-binding band structure calculations. Clearly, this is a major effect that affects the band structure and must be accounted for. After all, Sb monolayers are known to be 2D topological insulators thanks to this effect. The properties of the electronphonon interaction (viz., deformation potentials) are strongly affected by the spinorbit coupling.*

We are very grateful to the Reviewer for this comment. Indeed, in the original manuscript we did not include spinorbit coupling (SOC) effects, because we wanted mainly to focus on the conduction bands, whose relevant minimum is not degenerate and therefore not significantly affected by SOC. However, at the top of the valence band, the effects are important both in the band structure and in the electronphonon interaction as reflected by the deformation potentials (and actually it turns out

that SOC improves the mobility results, as discussed in the revised paper). Therefore, we have decided to perform from the beginning all simulations in the manuscript, including spinorbit effects. The main paper shows now only results including SOC, and we have moved all results without SOC and their comparison with SOC results in the Supplementary Information.

3. *The electronphonon matrix element exhibits a strong anisotropy in many cases. It is not clear how this has been accounted for by the authors who consider simple deformation potentials obtained from the applied strain rather than from a direct calculation of the electronphonon matrix elements, as it should be done.*

Arsenene and antimonene are isotropic materials owing to the underlying hexagonal pointgroup. This is the reason why mobilities are identical (within numerical accuracy) along both armchair and zigzag directions (note that, as explained in detail in the Supplementary Material, while singlevalley properties are anisotropic, the total mobility tensor has to be isotropic). The method we are using would anyway be able to capture anisotropic effects as it has been shown for instance in [J. Qiao et al., Nat. Commun. 5, 4475 (2014)] in the case of phosphorene.

4. *The "Takagi" formula for the carrier mobility is notoriously oversimplified. It has been used previously (see Shao et al., J. Appl. Phys. vol. 114, 093712 (2013) for the case of silicene), but it neglects important wavefunctionoverlap, degeneracy, and bandstructure effects. See also van de Broek et al., 2D Mat. vol. 3, 015001 (2016) for Sn, Si nanoribbons, for example.*

Indeed, while original Takagi's formula is limited to a single, nondegenerate band, the formula we are using here is its extension for a multivalley, anisotropic case, (see sentence in the paper after the formula: "in our case, we need to consider this formula in the multivalley, anisotropic case: details can be found in the Supplementary Information"; in the Supplementary Material we have explicitly derived the expression that takes into account degeneracies, bandstructure effects, and the lower (2D) dimensionality of the system). In fact, we are also citing not only Takagi's paper, but also [J. Qiao et al., Nat. Commun. 5, 4475 (2014)] and [J. Xi, Nanoscale 4, 4348 (2012)] where the formula was partially rederived to take into account some of these effects. In any case, even with these extensions, we acknowledge that some effects may not be fully captured by the formula. We want to also emphasize, though, that the aim of our paper is mainly to provide an assessment of the materials for electronics applications, as well as to evaluate the full device performances (in the later section). For this reason, in the paper, we mention that "the values that we calculate should be considered as upper limits to the actual carrier mobility" and we now add a sentence: "While accurate values for the electronphonon scattering terms can be obtained fully abinitio [S. Y. Savrasov et al., Phys. Rev. Lett. 72, 372 (1994); F. Giustino et al., Phys. Rev. B 76, 165108 (2007)], an efficient method to get satisfactory estimates relies on deformationpotential theory", wherein we also add some references to the relevant abinitio methods.

Also, to make the comparison with other materials more fair, we have now adapted the manuscript to explicitly distinguish the comparison with experimental results and with other "values predicted by simulations at the same level of theory", as e.g. the Nat. Comm. paper by J. Qiao et al., Nat. Commun. 5, 4475 (2014).

5. *The use of Wannier functions to "calibrate" tightbinding models has been described and used recently by the Purdue group (Kubis, Klimeck et al.).*

Indeed, the use of Wannier functions to extract tightbinding Hamiltonians for many different materials has seen a lot of applications in the recent literature. All the seminal papers on the approach are now cited in the paper: Souza, Marzari, Vanderbilt, PRB 65, 035109 (2001) introducing the method and the use of Wannier functions as a tightbinding basis; Lee, Nardelli, Marzari, PRL 95, 076804 (2005) for a first application of Wannier functions as a tightbinding basis in a multiscale approach; and Marzari et al., Rev. Mod. Phys 84, 1419 (2012) for a review of the method and its applications in the literature. We have also added a sentence and two references to other methods to obtain tightbinding models from DFT calculations, including the work by Kubis and Klimeck: “A few different methods have been proposed in the literature to address the issue of bridging the different simulation scales discussed above [Porezag et al, Phys. Rev. B 51, 12947 (1995); Tan et al., J. Comp. Electron. 12, 56 (2013)]. In this paper we have adopted...”

Reviewer #3

1. *It is left to at authors' discretion to add a paragraph on how the 2D arsenene and antimonene based nMOSFET can be realized experimentally.*

Indeed the experimental interest in arsenene and antimonene is growing fast, encouraged by theoretical papers on the promising properties of these 2D materials. Fewlayer systems have already been synthesized [e.g. T. Lei, J. Appl. Phys. 119 (2016)] and available technologies for 2D materials could be applied to realize the devices we put forward. We followed the Reviewer's suggestion and added a sentence in the introduction: “While this suggestion is reasonable, only an accurate simulation of a complete device can support this hypothesis and will further stimulate experimental interest [T. Lei, J. Appl. Phys. 119 (2016)] in these novel 2D materials.”.

2. *Some of long paragraphs in Introduction and Results & Discussion section are too long. It is advised to split in small paragraphs so that work will read more focused.*

We have revised the Introduction and Results & Discussion sections to shorten some sentences, and we have split the content in more paragraphs to ease their readability.

Reviewers' comments:

Reviewer #1 (Remarks to the Author):

All of the response are satisfactory.
I can recommend this paper to Nature Communications.

Reviewer #2 (Remarks to the Author):

I appreciate the efforts done by the authors to address my previous comments. In particular, I am now satisfied with the way they have addressed the issue of the spin-orbit coupling.

Having said that, I must reiterate the fact that buckled Sb and As monolayer (the most energetically favored 2D crystalline structure of these materials, in comparison with planar and puckered structures) allow first-order coupling with out-of-plane flexural acoustic modes (ZA phonons). Although these materials do not exhibit a Dirac-like electron dispersion, this coupling remain potentially extremely strong and can affect significantly the carrier mobility. In their

rebuttal letter, the authors claim that the phonon dispersion is renormalized to such an extent as to render this process negligible. I remain unconvinced: In the most thoroughly studied case, graphene, for example, this renormalization (mainly due to the anharmonic coupling between out-of-plane and in-plane acoustic modes) is effective, since it depresses the much more singular two-phonon process. To my knowledge, this has not been shown to be the case for antimonene and arsenene. Neither is this shown here. Also, whereas some renormalization (or phonon "stiffening") may be due to the mechanical coupling of the 2D layer to the gate and substrate material(s), I do not see any convincing and qualitative proof of the authors' claim.

I also remain unconvinced by the method used here to calculate the electron mobility: Use of the elastic constants makes it impossible to deal with the strong anisotropy of the electron-phonon matrix elements. In their rebuttal letter, the authors correctly emphasize the isotropy of the electron dispersion. However, this is not the (an)isotropy I had in mind in my third comment: The enhancement of backwards or forwards scattering (or whatever different angular dependence one may find as a result of the angular dependence of the phonon polarization vectors, of the electronic wavefunctions, and of their overlap integrals) will affect the mobility. This effect is completely ignored here.

In summary: I remain unconvinced. This is not to say that the manuscript presents results that are necessarily incorrect. Rather, the authors fail to consider several serious effects and do not provide any proof or justification for this. I think that a "sound" paper should present such proof or, at the very least, comment on the approximations made and estimate the possible sensitivity of the final results on these approximations.

Regarding a final recommendation, I think that this paper contains some very useful information and presents a very sound formulation of the problem. However, the shortcuts made to simplify the physical models and the possible (or even likely) importance of effects that have been ignored forces me to be extremely reluctant about recommending it for publication in its present form.

Reviewer #3 (Remarks to the Author):

Authors have attempted to include reviewers's comments and feedback into the revised version and supplemented with the supplemental material. The revised paper in the present form could be acceptable for publication.

Detailed answer to the comments of referee #2

Buckled Sb and As monolayer (the most energetically favored 2D crystalline structure of these materials, in comparison with planar and puckered structures) allow first-order coupling with out-of-plane flexural acoustic modes (ZA phonons). Although these materials do not exhibit a Dirac-like electron dispersion, this coupling remain potentially extremely strong and can affect significantly the carrier mobility.

We agree with the referee that the coupling of electrons with ZA phonons can be important in structures without planar symmetry, as we already stated in our manuscript (“out-of-plane (ZA) phonons may play an important role in free-standing Dirac materials without planar symmetry”), and that the Takagi’s formula does not capture such coupling.

To obtain an estimate of the relative importance of ZA phonons with respect to LA phonons, we have performed extensive additional calculations. Here are the details of the calculation:

- Code: EPW v. 4.0, released only a few weeks ago - in the previous versions, spin-orbit coupling was not supported (<http://epw.org.uk>, <http://arxiv.org/abs/1604.03525>);
- Electron k-mesh: 14x14x1; phonon q-mesh: 7x7x1; same cutoffs as the calculations used in the main paper;
- Interpolation on a dense 150x150x1 q-mesh via Wannier interpolation (same Wannier parameters used in the paper).

We would also like to stress that the calculations to obtain the results shown below have required a significant amount of time, in addition to modifications of the source code of EPW for the required $|g|^2$ coupling coefficients; moreover, we had to write the code for the post-processing, analysis and plotting of the results. Besides, all calculations have required ~ 8 CPU days of running time (with 16 parallel cores) to obtain the final quantities.

In the following plot we show, for 7 selected k-points, the *inverse lifetime* (as calculated by EPW) of some relevant valence (top row) or conduction (bottom row) electronic states, due to scattering with all phonons at any q vector of a given mode index (x axis). The three bars correspond to the following acoustic phonons: 1=ZA, 2=TA, 3=LA.

In particular, in the first row, the first plot (“valence band maximum”) refers to $\mathbf{k}=\Gamma$, the second plot (“inside Fermi sphere”) is at $\mathbf{k}=(0.0412,0,0)$ [in reciprocal lattice units], and the third plot at $\mathbf{k}=(0.0825,0,0)$.

In the second row, the first plot of the second row (“conduction band maximum”) refers to $\mathbf{k}=(0.31063,0,0)$, i.e. the minimum of the conduction band, and the other plots to neighboring \mathbf{k} -points chosen such that the respective band energy is within or outside the Fermi sphere (we have used the same n - or p -doping of the devices simulated in the main text). In particular the coordinates are: for the second plot, $\mathbf{k}=(0.3299,0,0)$, for the third plot $\mathbf{k}=(0.2474,0,0)$, and for the fourth plot $\mathbf{k}=(0.4124,0,0)$.

We note two behaviors:

- For *valence* states, the ZA coupling is always smaller than the LA coupling. Even near the Fermi energy, the *total* lifetime (ZA+LA+TA) is roughly twice the lifetime due to LA modes only. Therefore, we expect that our estimates in the valence can be overestimated by a factor of ~ 2 .
- For *conduction* states, instead, the ZA coupling is a bit stronger. We can estimate that the strength is approximately $ZA \sim 3LA$, $TA \sim LA$, so that we may be overestimating the conduction mobility by a factor of $\sim (3+1+1)=5$.

This is of course only a rough estimate, but it gives already the idea of what would change if a full calculation were performed. In addition, by taking into account also scattering with optical phonons and thus the contribution of inter-valley scattering for conduction states, we have evaluated that the mobility could be overall reduced up to a factor of 8.

Nonetheless, we stress that a complete calculation of the mobility would require the estimation (and then integration) of the couplings on a *dense* k -grid (say at least $\sim 50 \times 50$), with an increase in computational time of over three orders of magnitude (still affordable, but requiring months on supercomputers or large clusters).

We have adapted the main paper to mention that the effects that we are neglecting may lead to corrections up to a factor of 8: “*We have estimated that, in the worst case scenario, the values of actual mobilities could be reduced up to a factor of 8 when a full treatment of the electron-phonon coupling is adopted, including intervalley scattering.*”. Moreover, as stated in the next answers, this factor is going to be significantly reduced if the 2D channel material is deposited on a substrate or if a gate is present.

In their rebuttal letter, the authors claim that the phonon dispersion is renormalized to such an extent as to render this process negligible. I remain unconvinced: In the most thoroughly studied case, graphene, for example, this renormalization (mainly due to the anharmonic coupling between out-of-plane and in-plane acoustic modes) is effective, since it depresses the much more singular two-phonon process. To my knowledge, this has not been shown to be the case for antimonene and arsenene.

In our manuscript we did not refer to this kind of renormalization, but rather to a “phonon stiffening” (mentioned by the author in the next comment, see below).

Also, whereas some renormalization (or phonon “stiffening”) may be due to the mechanical coupling of the 2D layer to the gate and substrate material(s), I do not see any convincing and qualitative proof of the authors’ claim.

The phonon stiffening effect has already been shown to be effective is

Figure 3 The phonon spectra of single-layer and bilayer (a) graphene and (b) h-BN. For h-BN a Gaussian smearing and a non-analytical correction were used to account for the LO-TO splitting at $q \rightarrow \Gamma$ (see e.g. Ref. [48]). For both bi-layer systems we used the energetically most favourable stackings (AB for graphene and AA' for h-BN).

other 2D materials. As an example, in the following paper: *G. J. Slotman et al., Ann. Phys. (Berlin) 526, 381–386 (2014)* the authors show in Fig. 3 (reported also here on the right) the effect of (ZA) phonon stiffening for BN and graphene double layers. In that paper, it is clear that ZA phonons in the double layers (dashed blue lines) move at finite energies near Γ as a consequence of the interaction with a neighboring layer (we also emphasize that the interaction between layers in a bilayer is still van der Waals and doesn't have a strong chemical bonding character). The same effect will happen also in the case of As or Sb monolayers deposited on a different material, as e.g. the gate material or a substrate.

It is important to note that such an effect will strongly reduce the electron-phonon interaction of ZA phonons with respect to TA and LA phonons at room temperature, due to the change in Bose-Einstein occupation. In fact, the expression for the inverse lifetime $1/\tau_k$ contains a sum of terms of the type (e.g. in the case of phonon absorption) is:

$$\sum_{k',q} |\langle k' | \partial_q V | k \rangle|^2 n_q \delta(E_k - E_{k'} + \hbar\omega_q) \delta_{k+q, k'+G}$$

where n_q is the Bose-Einstein occupation factor for phonons (for simplicity, we are not indicating the electronic occupation that is also present in the formulas we used for the generation of all results presented here, see complete formula below). From our calculations using EPW, we have verified that if we don't include the phonon population n_q in the expression above, the final result is reduced by a factor of 20 for ZA phonons (e.g. in the case of the conduction band minimum), while it is reduced only by a factor of 5 for LA phonons. Therefore, the strong coupling of ZA phonons is mainly due to the very large Bose-Einstein occupation near Γ (exponentially increasing for small phonon frequencies ω_q). The phonon stiffening shown in the paper mentioned above will significantly depress the ZA phonon population and render the scattering with ZA phonons much less relevant in realistic devices.

Use of the elastic constants makes it impossible to deal with the strong anisotropy of the electron-phonon matrix elements. In their rebuttal letter, the authors correctly emphasize the isotropy of the electron dispersion. However, this is not the (an)isotropy I had in mind in my third comment: The enhancement of backwards or forwards scattering (or whatever different angular dependence one may find as a result of the angular dependence of the phonon polarization vectors, of the electronic wavefunctions, and of their overlap integrals) will affect the mobility. This effect is completely ignored here.

We agree that the Takagi's formula is a simple model and some effects may not be taken into account. Of course, an accurate numerical result can be obtained only with a full explicit calculation, which is however well beyond the scope of the present paper as already discussed. We show anyway some preliminary results obtained from the ab-initio EPW simulations that we performed, in order to reply to the referee.

We start showing a map of the electron-phonon coupling coefficients $|g_{k,q}|^2 = |\langle k+q | \partial_q V | k \rangle|^2$ as a function of q in the first Brillouin zone for the LA mode ("mode 3"). The next plots refer to the top valence band. The left panel is for $k=\Gamma$, while the right panel is the same plot for a k -point a bit outside of Γ (shifted along the vertical direction in the plots), chosen so as to be located near the Fermi surface.

It is clear that the left plot preserves the (hexagonal) symmetry of the system, while the plot on the right has only the symmetry of the small group of \mathbf{k} (mirror symmetry along a vertical axis). Before further discussing the isotropy of such curves, we remind that the $|g|^2$ coefficients have to be multiplied by the energy δ and the occupation factors. We show therefore in the following plots the quantity

$$|g_{\mathbf{k},\mathbf{q}}|^2 (n_{\mathbf{q}} + f_{\mathbf{k}+\mathbf{q}}) \delta(E_{\mathbf{k}} - E_{\mathbf{k}'} + \hbar\omega_{\mathbf{q}})$$

as a function of \mathbf{q} in the Brillouin zone, and for the same \mathbf{k} points of the previous figure:

It's clear that for $\mathbf{k}=\Gamma$ (left panel), only \mathbf{q} -points very near Γ contribute, and we can safely assume that the value is isotropic. Outside Γ (right panel) the distribution has an almost circular shape, with the circle touching the origin and with the center displaced in the direction of the chosen \mathbf{k} point. The shape is not isotropic, as expected, but we have to remember that in order to get the mobility we need to sum on all \mathbf{k} points, which will therefore average to give an isotropic result.

In a similar fashion, the LA $|g|^2$ couplings with an electron at the bottom of the conduction band, shown in the figure below, are not isotropic: this is expected, as stated in the paper, because we are looking at a single valley. But also in this case, after averaging on all the \mathbf{k} points in the valley, and on all 6 valleys, the final mobility will be isotropic.

Even in spite of all these considerations, we are aware that a correct numerical evaluation of the mobility needs a calculation considering explicitly the distribution of $|g|^2$ that we have calculated in the plots above. However, as already explained, this task is quite difficult and beyond the scope of the paper.

I think that a "sound" paper should present such proof or, at the very least, comment on the approximations made and estimate the possible sensitivity of the final results on these approximations.

As the referee suggests, we have improved the manuscript text to emphasize the approximations made, what are the possible limitations, and to estimate the sensitivity of the final results on these approximations. In particular, we now mention even more clearly that “*we will limit the analysis only to the intrinsic in-plane mobility intrinsically limited by the scattering with longitudinal acoustic (LA) phonons*” and that “*the values that we calculate should be considered as upper limits to the actual scattering times or, equivalently, to the carrier mobility*”.

More importantly, we added a paragraph to discuss the limitation of the approximation and the estimate of the sensitivity of the results: “*We would like to emphasise, however, that this formula, while often adopted in the literature, cannot be used to obtain a quantitative estimate of the mobility. Indeed, the formula neglects the coupling with ZA phonons (which may be important, as already discussed above), as well as with TA and optical phonons. Moreover, it cannot fully capture the anisotropy of the electron–phonon coupling coefficients. A full ab-initio treatment of the electron–phonon scattering is required if a quantitative estimation is required (see e.g. discussions in Refs. 31 and 39). Nevertheless, we provide here an estimate of what we will call hereafter “Takagi’s mobilities”, mainly to allow to compare As and Sb with other 2D materials already investigated in the literature within the same level of theory. We have estimated that, in the worst case scenario, the values of actual mobilities could be reduced up to a factor of 8 when a full treatment of the electron-phonon coupling is adopted, including intervalley scattering*”.

In the remaining of the paper, we always refer to the “Takagi’s mobility” to avoid confusion. Finally, in the conclusion, we state that “*our single-valley and multi-valley upper estimates of the mobilities in the Takagi’s approximation show that high phonon-limited mobilities can potentially be obtained both [...] even if ab-initio simulations of the electron–phonon scattering are required to obtain quantitative predictions for this quantity.*”

We have decided to keep anyway the estimation of the mobility using Takagi’s formula in our manuscript, because it allows to compare our results with other literature work on different materials, evaluated using the same approximation, like the following ones for instance: *J. Qiao et al., Nature communications 5, 4475 (2014); R. Fei, L. Yang, Nano Letters 14, 2884 (2014); H.M. Stewart et al., Nano Letters 15, 2006 (2015); G. Schusteritsch, Nano Letters 16, 2975 (2016), M.Q. Long, J. Am. Chem. Soc. 131, 17728 (2009); W. Zhang, Nano Research 7, 1731 (2014).*

We also attach a version of the paper with all modification highlighted, where also some other minor changes have been applied in line with the modifications discussed here.